# Particle-resolved topological defects of smectic colloidal liquid crystals in extreme confinement

René Wittmann [1,4 ✉], Louis B. G. Cortes [2,3,4], Hartmut Löwen[1 ✉] & Dirk G. A. L. Aarts[2 ✉]

Confined samples of liquid crystals are characterized by a variety of topological defects and can be exposed to external constraints such as extreme confinements with nontrivial topology. Here we explore the intrinsic structure of smectic colloidal layers dictated by the interplay between entropy and an imposed external topology. Considering an annular confinement as a basic example, a plethora of competing states is found with nontrivial defect structures ranging from laminar states to multiple smectic domains and arrays of edge dislocations, which we refer to as Shubnikov states in formal analogy to the characteristic of type-II superconductors. Our particle-resolved results, gained by a combination of real-space microscopy of thermal colloidal rods and fundamental-measure-based density functional theory of hard anisotropic bodies, agree on a quantitative level.

[1] Institut für Theoretische Physik II: Weiche Materie, Heinrich-Heine-Universität Düsseldorf, Düsseldorf, Germany. [2] Department of Chemistry, Physical and Theoretical Chemistry Laboratory, University of Oxford, Oxford, UK. [3] School of Applied and Engineering Physics, Cornell University, Ithaca, NY 14853, USA. [4] These authors contributed equally: René Wittmann, Louis B. G. Cortes. ✉email: rene.wittmann@hhu.de; Hartmut.Loewen@uni-duesseldorf.de; dirk.aarts@chem.ox.ac.uk

L iquid crystals consist of particles that possess both translational and orientational degrees of freedom and exhibit a wealth of mesophases with partial orientational or positional order such as nematic, smectic and columnar states[1]. As such, these phases are highly susceptible to external topological and geometrical influences[2]. This opens a fascinating new research realm on the internal structural response to such externally imposed constraints with various highly relevant applications in technology and material science[3,4]. While these perspectives have been extensively exploited for spatially homogeneous mesophases, such as nematics, there is much yet undisclosed potential stemming from the complex interplay between external constraints and internal order emerging in more complex mesophases, such as the layered smectic phase.

One of the main research goals in liquid crystals is focused on topological defects. These not only represent fingerprints of singularities and discontinuities in the ordering but also naturally link topology to condensed matter physics. The general importance of defects of liquid crystals is further fueled by the possibility to directly visualize the inherent orientational frustration on the macroscopic scale through the schlieren texture between two crossed polarizers[1]. Different orientational defect structures have therefore been explored a lot in the homogeneous nematic phase[5–19] with recent digressions to active systems[20–22]. Due to their simultaneous orientational and positional ordering, defects in the smectic phase naturally exhibit an even higher degree of complexity. The main emphasis has been put hitherto on the positional layering[23–27] or orientational textures[19,28,29] alone, as well as, on coarse-grained calculations[1,19,25,29–32] and computer simulation of particle models[33–36].

Here we approach the defect structure of smectic liquid crystals from the most fundamental particle-resolved scale and quantify both their positional and orientational disorder simultaneously in theory and experiment. In doing so we study two-dimensional smectics composed of lyotropic colloidal rods whose size enables a direct observation[37–39], while they have the advantage over granulates[40,41] that they are fully thermally equilibrated. The colloidal samples are exposed to extreme confinements possessing an annular shape and dimensions of a few particle lengths. This combination of curved geometry and nontrivial topology is triggering certain characteristic defect patterns. In our study we uniquely combine real-space microscopy of colloidal samples with modern first-principles density functional theory (DFT)[42] based on geometric fundamental measures[43] which provides a full microscopic description of inhomogeneous and orientationally disordered smectics[44,45]. A plethora of different states with characteristic defect topologies is observed in perfect agreement between theory and experiment up to the microscopic nuances in the defect shape and wall alignment.

Our study explores the intriguing competition between the internal liquid crystal properties and the extrinsic topological and geometrical constraints. In annular confinement this gives rise to three essential types of smectic defect configurations. Each of these defects is characteristic for a unique state with a discrete rotational symmetry in the orientation field, which we refer to as follows. In the laminar states, the smectic layers in a large defect-free domain (bounded by two parallel disclination lines) resemble the flow lines around a circular obstacle (inclusion). The domain states are governed by individual smectic domains, separated by radially oriented disclination lines, in three sectors of the annulus. Finally, there are the Shubnikov states, named[30] in the formal analogy between the typical arrays of edge dislocations and the flux quantization in type-II superconductors. In addition, we observe peculiar symmetry-breaking composite states which unite different types of defects in a single structure. A locally adaptable layer spacing is found here to play the key

role regarding the stability and distribution of defect structures in extreme confinement.

## Results

**Overview**. Our complementary experimental and theoretical strategies (see methods section and Supplementary Note 1 for more details) to study smectic liquid crystals on the particle scale are illustrated in Fig. 1. Experimentally, we directly observe fully equilibrated silica rods at the bottom of customized confinement chambers (Fig. 1a) through particle-resolved bright-field microscopy (Fig. 1b). On the theoretical side, we analyze the microscopic density profiles, obtained from a free minimization of our geometrical DFT for two-dimensional hard discorectangles (Fig. 1c). We strive a direct comparison of experimental snapshots and theoretical density profiles, as illustrated for the bulk case in Fig. 1d, where the DFT is minimized by the optimal layer spacing $\lambda_0$ (Fig. 1e). To create the required overlapping parameter space, we employ both precise lithography to create robust confinement chambers with dimensions of only a few rodlengths and an efficient hard-rod density functional to tackle these system sizes. The theoretical aspect ratio $p = 10$ is chosen to closely match the effective value $p_{\text{eff}} = 10.6$ in the experiment and the density of rods is chosen to be slightly above the bulk nematic–smectic transition in each case.

The main results of our joint experimental and theoretical study of colloidal liquid crystals in annular confinement (see Fig. 1b for the relevant geometrical parameters) are summarized as follows. First, we identify a plethora of distinct smectic states (laminar, domain, Shubnikov and composite), shown in Fig. 2, which possess a unique defect structure, layer arrangement and director topology. Second, we predict in Fig. 3 a transition from the laminar state for small inclusion sizes to the Shubnikov state for large inclusion sizes, which nonmonotonically depends on the total confinement size. Third, analyzing the characteristic microscopic (Fig. 4) and topological (Fig. 5) details of each state explains the stability of the observed structures. Finally, we illustrate in Fig. 6 the full microscopic variety of different structures, including a stable composite state. We further argue with the aid of Fig. 7 how the state diagram changes with varying density and rod length.

**Smectic states**. Figure 2 illustrates our central observation of different competing smectic states, each coping with the externally imposed constraints in a distinct way. For each experimental structure in a given geometry, we find a perfectly matching theoretical density profile. This depicted structural variety results from a confinement with curved walls and a nontrivial topology, represented here by an annular cavity (Fig. 1b) with Euler characteristic $\chi = 0$. The typical structure of these states is determined by the arrangement of the smectic layers (see microscopic details below) and the shape of topological defects with total charge of $Q = \chi = 0$ (see topological details below). A detailed classification of the observed smectic states is given in Supplementary Note 2.

To demonstrate the plain behavior of smectics in simply connected domains, we first remove the inclusion and consider circular confinement. In this reference case, the only structure observed is the bridge state ($\mathcal{B}$). It is characterized by a large domain of parallel layers spanning the system and frustration of the orientational order at the domain boundary. The latter is either due to the formation of two anti-radial disclination lines[34,39] or, if the layers are directly adjacent to the outer wall, due to homeotropic (perpendicular to the wall) alignment[33], contrasting the preferred planar alignment at hard walls.

When adding a small inclusion, the layer arrangement resembles a laminar flow field around an obstacle, which we refer

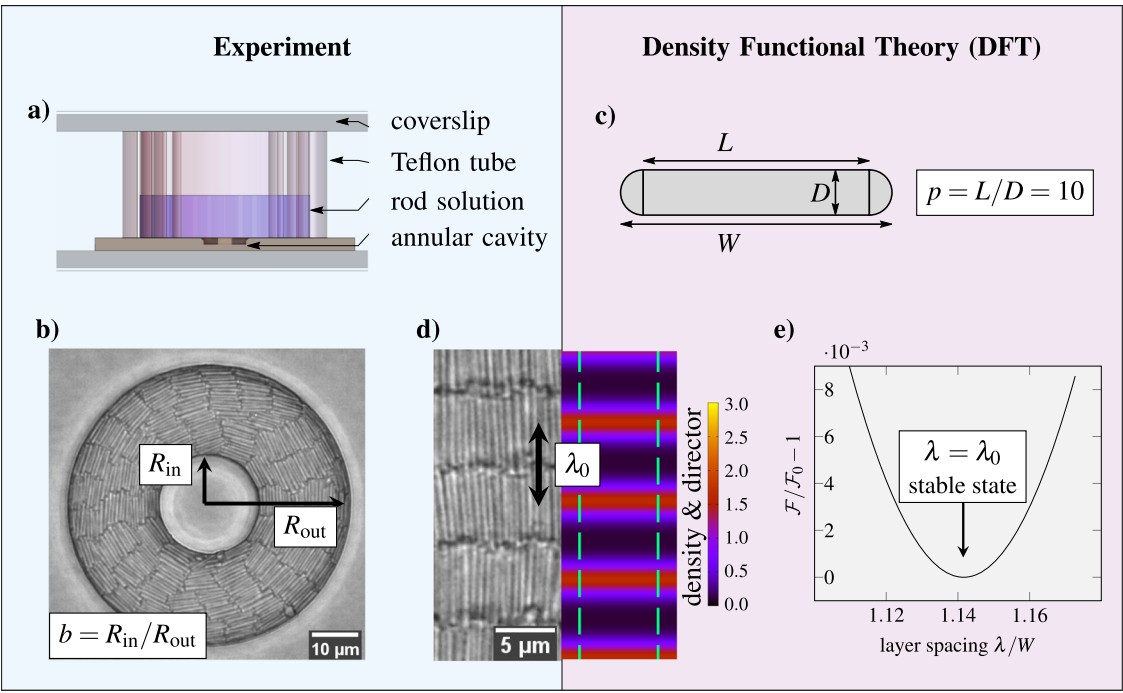

**Fig. 1 Overview of experimental and theoretical methods. a** Schematic illustration of the experimental cell. **b** Particle-resolved bright-field microscopy snapshot imaged in the direct vicinity of the cavity bottom wall. The annular geometry is determined by the outer radius $R_{out}$ and the inclusion size ratio $b = R_{in}/R_{out}$ with the inner radius $R_{in}$. **c** A discorectangle of rectangular length $L$, circular diameter $D$, total length $W = L + D$ and area $a = LD + D^2\pi/4$ used as the theoretical model particle. **d** Smectic bulk phase. Left: experimental snapshot showing individual particles. Right: theoretical density profile $\rho(\mathbf{r}, \phi)$ represented by a heat map of the orientationally averaged center-of-mass density $\bar{\rho} = \frac{a}{2\pi}\int_0^{2\pi} d\phi\, \rho(\mathbf{r}, \phi)$ and green arrows indicating the average local director orientation. The large black arrow marks the spacing $\lambda_0$ between two layers. **e** Bulk smectic free energy $\mathcal{F}$ as a function of the layer spacing $\lambda$, determined by density functional theory (DFT). The minimum at $\mathcal{F} = \mathcal{F}_0$ corresponds to the optimal bulk layer spacing $\lambda_0$ in equilibrium.

to as the laminar state ($\mathcal{L}$). The structure associated with the large bridging domain is identical to that of the bridge state, but the internal boundary may additionally disconnect some of the central layers and induce orientational frustration in the two tangential layers. The bridge state can thus be considered as a special undeformed case of a laminar state in the limit $b \to 0$.

At intermediate inclusion sizes, we observe a smectic domain state ($\mathcal{D}$) with three radially oriented disclination lines, exhibiting a characteristic zig-zag pattern on the particle scale.

Following de Gennes[30], we refer to the smectic structure at large inclusions as the Shubnikov state ($\mathcal{S}$). It is characterized by layers spanning between the two disconnected system boundaries and an array of edge dislocations, which stabilizes the uniform orientational bend deformation imposed here by the confining geometry. This structural response is mathematically analogous to the magnetic vortices emerging in superconductors of type II subject to an external magnetic field[1,30].

All smectic states introduced so far possess a discrete rotational symmetry. In addition, composite states ($\mathcal{C}^{\mathcal{LD}}$, $\mathcal{C}^{\mathcal{DS}}$ or $\mathcal{C}^{\mathcal{LS}}$) with two distinct regions, displaying characteristic order phenomena of either laminar, domain or Shubnikov states emerge at intermediate inclusion sizes. A key paradigmatic example shown in Fig. 2 is the laminar–Shubnikov composite state $\mathcal{C}^{\mathcal{LS}}$.

**State diagram.** To answer the question about the stability of each state, we illustrate in Fig. 3 the probability of its occurrence in our experiments. For all state points considered, laminar states and Shubnikov states clearly dominate for $b \lesssim 0.25$ and $b \gtrsim 0.35$, respectively. This provides compelling experimental evidence for the existence of a topological laminar–Shubnikov ($\mathcal{LS}$) transition around $b \approx 0.3$. The state diagram is complemented by the intermediate

domain state and several composite states. We observe that upon shrinking the system for a fixed intermediate inclusion size ratio $b \approx 0.3$, both the laminar state and the Shubnikov state become less stable, while the probability to find the domain state drops for larger systems.

The described state diagram can be qualitatively understood in terms of a minimalistic phenomenological model (see Supplementary Note 3), accounting solely for the length of the disclination lines ($\mathcal{L}$ and $\mathcal{D}$) or the number of edge dislocations ($\mathcal{S}$), cf. Fig. 2. Only in the domain state the length of the disclination lines depends on the inclusion size ratio $b$, such that the laminar–domain transition can be located at a fixed $b = 1/3$, independently of the unknown defect energy $\delta$ per unit length. The energy of the Shubnikov state also depends on the inclusion size ratio $b$, which is required to estimate for the total number of edge dislocations of energy $u_{ed}$. The model thus predicts an alternative laminar-Shubnikov transition, depending on the ratio of $u_{ed}$ and $\delta$. This fit parameter can be estimated by localizing the transition at the observed $b = 0.3$, which gives rise to the scenario depicted in Fig. 3a, where the domain state is only metastable.

To corroborate our experimental findings in full depth, we compute the free energies corresponding to the microscopic density profiles as a direct measure for their stability. Focusing on the precise localization of the $\mathcal{LS}$ transition, we observe in Fig. 3 a clear trend that the transition line shifts to larger inclusions for smaller $R_{out}$ in the experimentally accessible range of this parameter. This agrees well with the distribution of the observed structures in our experiments. For even more extreme confinements with $R_{out} \leq 2.1L$ we locate the $\mathcal{LS}$ transition close to the maximal inclusion size where a laminar state can form. Hence, the inclusion size ratio $b$ of the transition becomes smaller upon further decreasing $R_{out}$ below

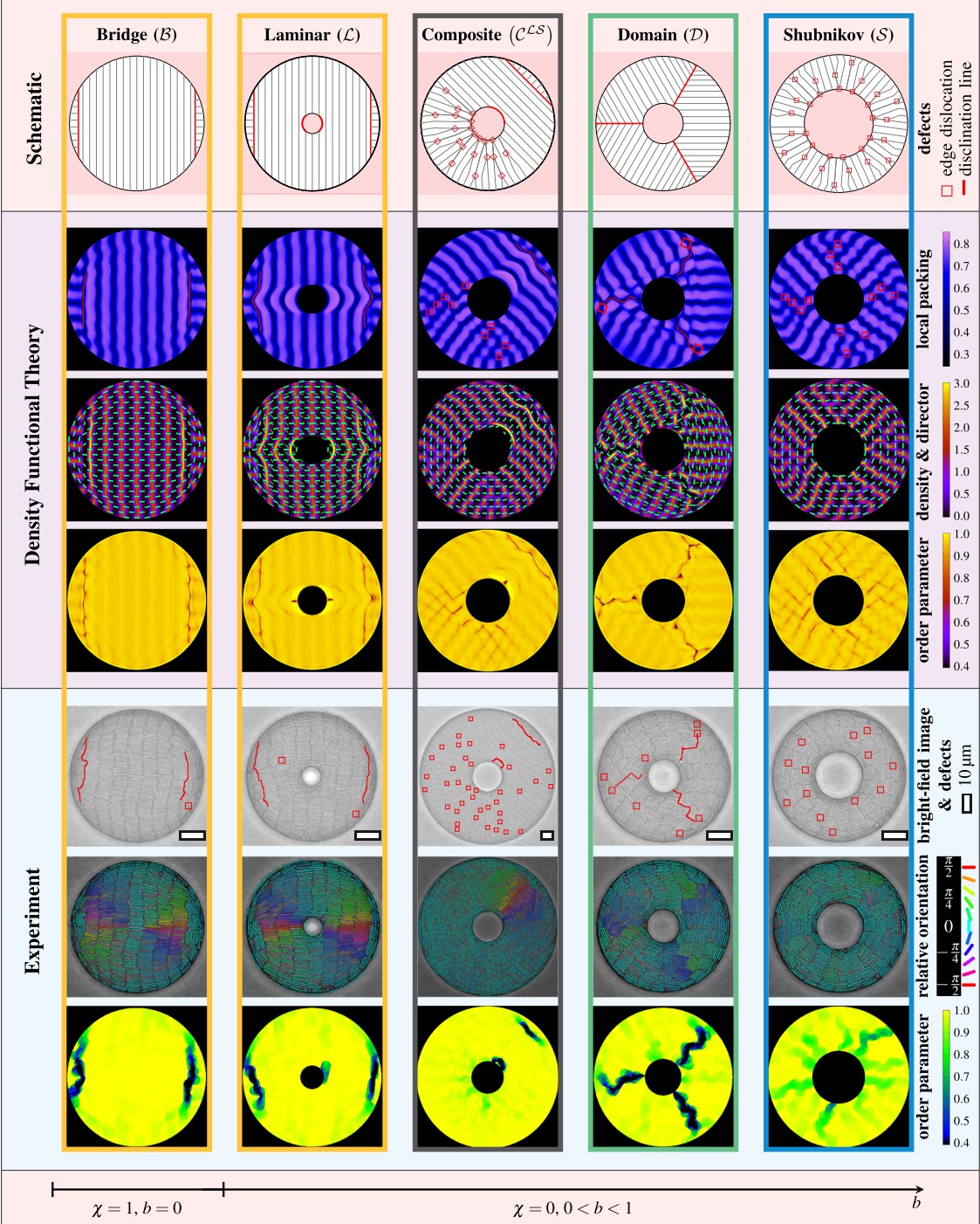

**Fig. 2 Defects structures in smectic colloidal liquid crystals.** The columns represent the different states (as labeled), arranged from left to right by their occurrence in circular (Euler characteristic $\chi = 1$) and annular ($\chi = 0$) confinement with increasing inclusion size ratio $b = R_{in}/R_{out}$. First row: idealized schematic representation of the mesoscopic arrangement of smectic layers (solid gray lines) and defects (red squares ☐ and lines ▬ as labeled). Rows 2–4: local packing fraction with marked defects, density profiles with orientational director field (as in Fig. 1d) and local order parameter from theory. Rows 5–7: typical particle-resolved snapshots with defects or color denoting the orientation relative to the nearest wall and local order parameter from experiment.

that threshold. For large confinements, the transition seems to approach the continuum limit with $b \approx 0.27$.

Further theoretical analysis shows that the composite state $\mathcal{C}^{\mathcal{LS}}$ is globally stable in a small but distinct region around the predicted $\mathcal{LS}$ transition, which we can understand on a microscopic level.

**Microscopic details**. The entropically optimal equilibrium structure of each state emerges from a complex balance between several competing driving forces which aim to (i) remove all sorts of defects, (ii) achieve planar wall alignment, (iii) minimize the deformation energy and (iv) maintain the intrinsic layer spacing $\lambda_0$ in bulk. Our particle-resolved methods naturally provide an

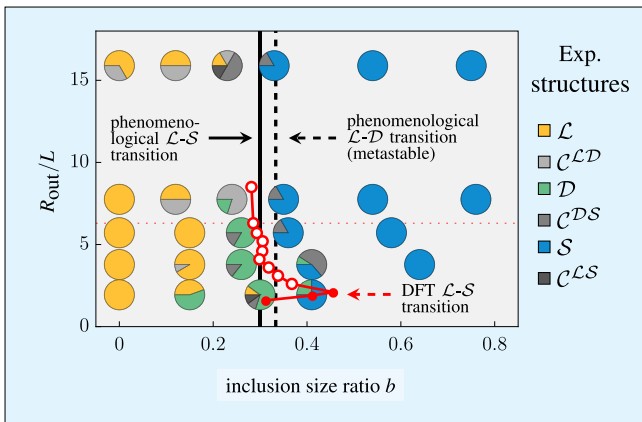

**Fig. 3 Topological state diagram.** Shown are the stable states for different outer radii $R_{out}$ and inclusion size ratios $b = R_{in}/R_{out}$. The large pie charts indicate the percentage at which each state occurs in the experiment according to the legend. The small circles denote the theoretical laminar-Shubnikov ($\mathcal{LS}$) transition (the numerical error is of the order of the symbol size; filled circles indicate that no laminar state can exist for larger inclusions). The vertical lines represent a possible scenario (see labels) predicted by a phenomenological model based on defect energies. Theoretical source data are provided as a Source Data file.

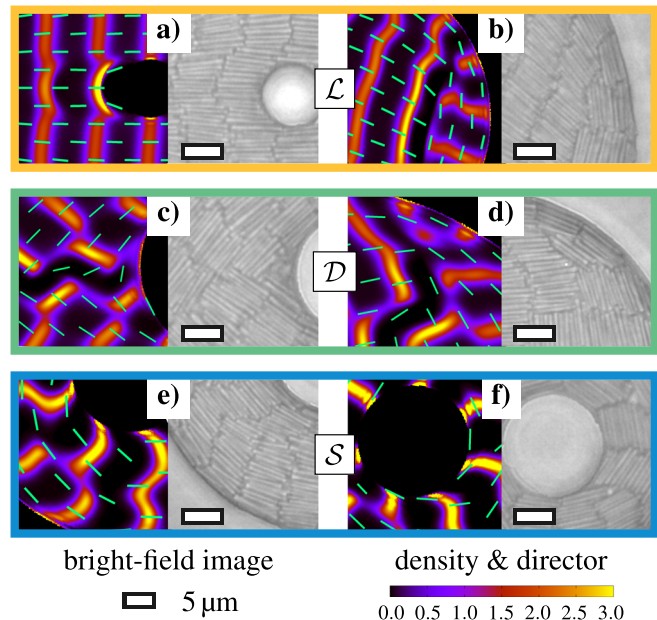

bright-field image          density & director

□ 5 μm          0.0 0.5 1.0 1.5 2.0 2.5 3.0

**Fig. 4 Structural details on the particle scale.** According observations from theory (left, density plots as in Fig. 1d) and experiment (right): **a** bent layers adjacent to the inclusion and **b** deformed layer near curved outer wall in the laminar state ($\mathcal{L}$), and **c** gap between two domains close to the inclusion in the domain state ($\mathcal{D}$) and **d** planar alignment of rods at the outer wall **e** deformed layers adjacent to an edge dislocation and **f** tilted alignment of some layers at the inclusion in the Shubnikov state ($\mathcal{S}$).

optimal account of points (i)–(iv) in the course of equilibration. The quantitative agreement of the experimental and theoretical density profiles, both generated by the subtle interplay of these fundamental principles, allows us to unveil in Fig. 4 the characteristic microscopic structural details of each state.

From our microscopic insights, detailed below and further elaborated in Supplementary Note 4, we draw the following conclusions regarding the state diagram in Fig. 3. Increasing the wall curvature (by decreasing $R_{out}$) increasingly distorts the layer spacing in the Shubnikov state, such that it becomes destabilized compared to the laminar state, for which the relative energy penalty arising from homeotropic wall alignment decreases. The nonmonotonic behavior of the resulting $\mathcal{LS}$ transition line is related to the varying compatibility of each state with the particular confining geometry. The indirect $\mathcal{LS}$ transition via an intermediate composite state $\mathcal{C}^{\mathcal{LS}}$ can be explained by the increased number of possibilities, compared to the $\mathcal{L}$ and $\mathcal{S}$ states, to relax the geometrical constraints.

In detail, we observe that the layers and defect lines surrounding the inclusion in the observed laminar states are typically deformed according to the shape of the inner wall, where the wall alignment is homeotropic, see Fig. 4a. In contrast, due to the planar alignment at the outer wall, the disclination lines end on point defects, recognizable by the modulation of the adjacent layer, see Fig. 4b. This has not been reported for straight walls[34,39]. The optimal numbers of layers depends nonmonotonically on the geometric parameters (see Supplementary Fig. 1).

The particularly deformed microstructure of the domain states is due to the competing angles $2\pi/3$ between the domain boundaries and $\pi/2$ between the intersecting layers. The positional order is most frustrated in the vicinity to the inclusion, as reflected by some dilute regions, shown in Fig. 4c, in which the rods try to align with the wall. Taking a closer look at the outer boundary, however, we observe in Fig. 4d some additional layers and edge dislocations between two adjacent domains, ensuring again an overall planar wall alignment.

The layers in the Shubnikov states are deformed in the vicinity of edge dislocations, see Fig. 4e. Although an overall planar wall alignment is generally possible, we frequently observe in small

systems that one or more layers are tilted with respect to the inner wall, as in Fig. 4f. This reflects a strongly position-dependent layer spacing (see Supplementary Fig. 2), which further distinguishes the Shubnikov from the laminar and domain states. In fact, the deviations in the local layer spacing reduce the number of point defects, such that we even observe some extreme structures without any defects for sufficiently small distances between the walls (see Supplementary Figs. 9 and 11 for some examples).

**Topological details**. Having resolved the microscopic details of the topological defects, emerging due to the rigidity of the smectic layers, we are in a position to associate in Fig. 5 a topological charge $q$ with each occurring disclination line. We further identify pairs of end-point defects to these lines, which formally carry peculiar quarter-integer charges $q_e$, such that $q = \Sigma q_e$. Then one can easily verify from the sketches in Fig. 2 that, in each state, the total charge $Q = \Sigma q$ equals the Euler characteristic of the confining domain, as required by topology[5,46]. The topological protection due to charge conservation is discussed in Supplementary Note 5.

The anti-radial disclination lines in the laminar (and $\mathcal{C}^{\mathcal{LS}}$ composite) state, can be understood as an expanded $q = +1/2$ point defect with two $q_e = +1/4$ charges at the end, see Fig. 5 (top). For the laminar structures without any disclination lines, there exists an equivalent $q = +1/2$ defect attached to the boundary, recognizable by the misalignment at the wall. Likewise, misalignment near the inclusion implies a negatively charged boundary defect with $q = -1/2$, see Fig. 5 (middle). Some structures (compare, e.g., Supplementary Fig. 7 or the experimental $\mathcal{C}^{\mathcal{LS}}$ in Fig. 2), display an explicit disclination line close to the inclusion ending on two $q_e = -1/4$ points.

The radial disclination lines are always in the interior of the system carrying the opposite end-point charges $q_e = -1/4$ and $q_e = +1/4$, close to the inner and outer end, respectively, see Fig. 5

| disclination line & layers | disclination line & director field: from nematic to smectic | end-point defects & director field |
|---|---|---|
| interior and boundary defect at outer wall | $q = +\frac{1}{2}$ | $q_e = +\frac{1}{4}$ <br> $q_e = +\frac{1}{4}$ |
| boundary and interior defect at inner wall | $q = -\frac{1}{2}$ | $q_e = -\frac{1}{4}$ <br> $q_e = -\frac{1}{4}$ |
| stretched and annihilated edge dislocation | $q = 0$ | $q_e = +\frac{1}{4}$ <br> $q_e = -\frac{1}{4}$ |

**Fig. 5 Topological defects in confined smectics.** The anti-radial line disclinations at the outer (top row) and inner (middle row) wall are located in the interior or attached to the boundary. The radial line disclinations are topologically equivalent to edge dislocations (bottom row) and can be interpreted as a stretched or annihilated defect, respectively. Left column: schematic illustration as in Fig. 2. Middle column: orientational director field (blue lines/arrows) around line defects (red) with topological charge $q$ according to the drawn closed integration path (cyan circular arrow). The disclination lines can be interpreted as stretched nematic point defects, shown for comparison. Right column: director field at the end-point defects (violet) of disclination lines with charges $q_e$. Here the integration path is not closed but rather begins and ends on two sides of the disclination.

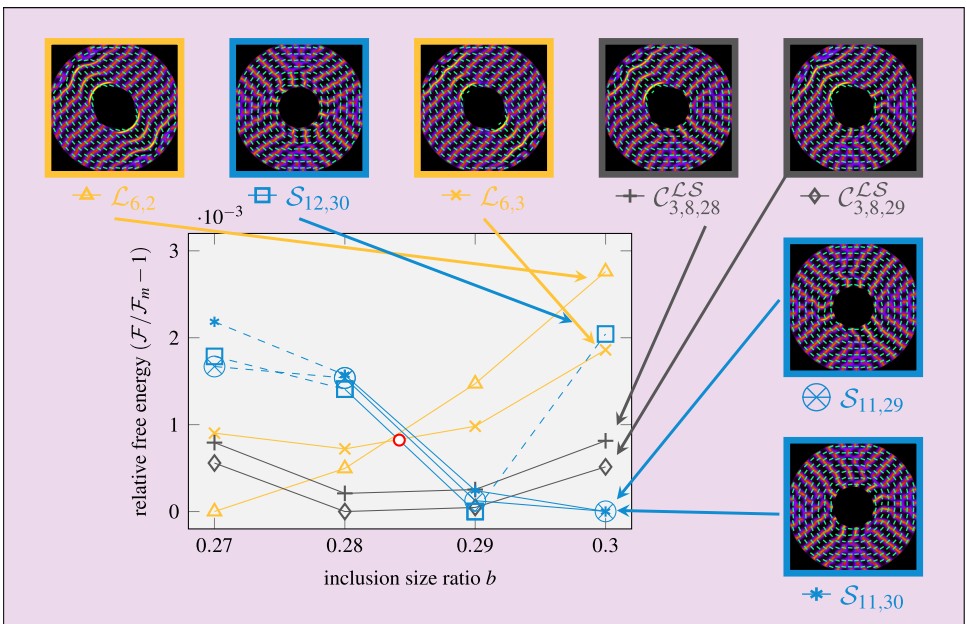

**Fig. 6 Stable and metastable structures for different inclusion sizes.** Shown is the free energy $\mathcal{F}$ relative to that $\mathcal{F}_m$ of the global minimum for $R_{out} = 6.3L$ and different inclusion size ratios $b$ close to the laminar-Shubnikov transition (red circle). The legend depicts the theoretical density profiles (as in Fig. 1d) of different states (color) with different microscopic structure (symbols) for $b = 0.3$. The structures with alike symbols are created by subsequently equilibrating the density for slightly smaller inclusions, where the solid lines serve as a guide to the eye and the dashed lines indicate a structural change regarding the number of layers in contact with the inclusion. The numerical error is of the order of the symbol size. Source data are provided as a Source Data file.

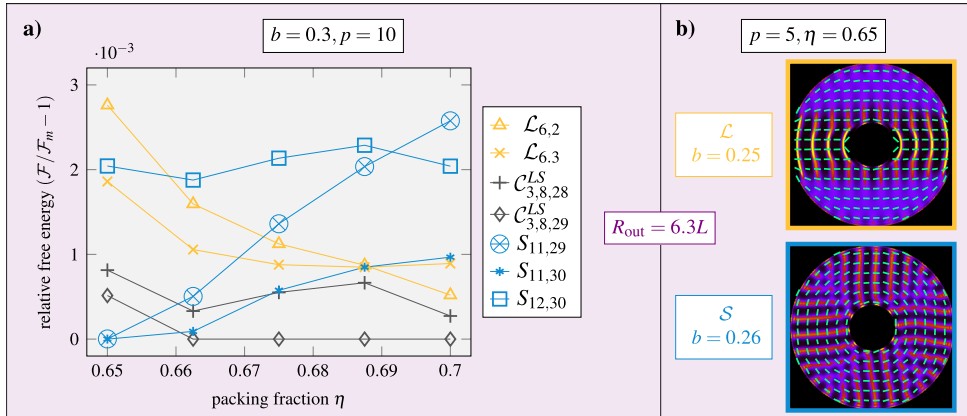

**Fig. 7 Stable and metastable structures for different intrinsic parameters.** The geometry is fixed by $R_{\text{out}} = 6.3L$ and $b = 0.3$. **a** Dependence of the relative free energy on the density. The data for area fraction $\eta = 0.65$ (at $b = 0.3$) and the estimated numerical error are the same as in Fig. 6. Here, alike symbols denote the structures obtained by gradually increasing the area fraction in steps of 0.0125. **b** Stable structures (as in Fig. 1d) for shorter rods with $p = 5$ with inclusion size ratios $b = 0.25$ and $b = 0.26$. The laminar-Shubnikov transition is thus located in between. Source data are provided as a Source Data file.

(bottom). Hence this type of line defect is nothing more than an expanded edge dislocation with topological charge $q = 0$, which unveils the true topological nature of the domain states. They possess the same orientational topology as the Shubnikov states into which they can evolve upon pair annihilation (compare Supplementary Fig. 3). The observation of domain states is thus owed to packing effects.

**Free energy landscape**. As apparent from the multitude of observed structures in some geometries, the system does not always equilibrate towards the global energy minimum. It is thus important to understand the full free energy landscape generated by the described competing driving forces. To this end, we additionally calculate the free energy associated with various theoretical density profiles, to directly assess their stability. As a representative example, we choose $R_{\text{out}} = 6.3L$ and compare in Fig. 6 seven sets of structures obtained by smoothly decreasing $b$ in the vicinity of the laminar–Shubnikov transition. We draw four important conclusions.

First, we explicitly see that the laminar–Shubnikov transition is not sharp. Instead, over a significant range $\Delta b \approx 0.015$ of inclusion size ratios, a composite state of both structures is energetically favorable. Second, the free-energy differences between two distinct structures are extremely small and the optimal microscopic structure changes multiple times upon small modifications of the confinement. These observations explain the large number of different structures observed in the experiment for $b \approx 0.3$. Third, it is important to identify the optimal microscopic structure of each state to make a proper statement about possible topological transitions. For example, only considering for $b = 0.3$ the metastable Shubnikov structure with the largest energy would lead to the false conclusion that the laminar state (or a composite state) is more stable in this geometry. Finally, a smooth variation of the inclusion size leaves laminar structures invariant (topologically protected), while the Shubnikov structures gradually adapt to the change in geometry, sometimes following a small hysteresis loop (see Supplementary Note 6).

**Dependence on density and rod length**. Apart from the external topological and geometrical constraints, the formation and stability of the reported states also depend on different intrinsic parameters (see Supplementary Note 7 for more details). The effect of the preferred bulk layer spacing $\lambda_0$ is illustrated in Fig. 7.

We see that increasing the density (Fig. 7a), resulting in a smaller $\lambda_0$, stabilizes laminar structures compared to Shubnikov structures, while decreasing the aspect ratio to $p = 5$ (Fig. 7b), resulting in a larger relative $\lambda_0/p$, has the opposite effect. Extending our state diagram towards shorter rods at a fixed density, we further anticipate the emergence of stable tetratic structures[41], since smectic order is generally destabilized[47].

For long rods at lower densities, different nematic states $D_n$ are found[9], classified by the number $n$ of $q = \pm 1/2$ defect pairs. From a topological point of view, the laminar and Shubnikov states can be interpreted as the smectic analogy to $D_2$ and $D_\infty$, respectively, while possessing a distinct orientational director field (compare Fig. 5), imprinted by the arrangement of smectic layers. The smectic analogy to $D_3$ (three-line disclination of charge $q = +1/2$ close to the outer wall) is not stable here as the curved geometry requires too strong deformations (see Supplementary Fig. 4), while there is no direct nematic analogy to domain and composite states. In our experiment, we observe the formation of nematic states at the bottom of our chambers in course of the sedimentation process (see Supplementary Movie 1), which can be mimicked in DFT by subsequently increasing the density. From the latter approach, we predict that the onset of smectic order depends on the underlying nematic state (see Supplementary Fig. 5). In turn, the nematic order in sedimentation equilibrium is presumably dictated by the more rigid smectic structure observed at the bottom.

## Discussion

We have performed a complementary particle-resolved experimental and theoretical study of hard colloidal rods in annular confinement. Our observations emerge from the fundamental principle of globally maximizing the entropy subject to the constraints arising from the external influences of the confinement and the internal smectic layer structure, which depends on the particle shape and density. All these competing driving forces are accounted for explicitly by our DFT data for the equilibrated structures.

In the future, it will be interesting to have a closer look at the position dependence of particle diffusion between the layers[48] or the formation dynamics of the different smectic structures, e.g., using dynamical DFT[49,50]. Drawing phase-stacking diagrams[51] will provide vital information on how the coexistence of nematic and smectic structures affects their stability in the experiment. While some additional smectic states could become stable in

different geometries, an even larger structural variety can be anticipated in more complex topologies, e.g., those with two holes. The next level of geometrical and topological complexity will be reached when immersing colloidal smectic liquid crystals in random porous media[52–55] and fractal confinement[56]. On the other hand, there is also a high intrinsic potential for finding novel structures when considering more exotic particle shapes[57–61].

In conclusion, we have shown that the topology and geometry of an externally imposed confinement largely determine the preferred internal structure of a smectic liquid crystal. Adjusting these screws allows to create a protocol for a guided self-assembly of a desired defect structure. Owing to their robustness and large range of metastability, the described smectic states can then be smoothly transferred to any desired confining geometry and, if desired, solidified to unfold their potential for various micro-technological applications[3,4]. These possibly include novel devices for information storage, templates for functional microstructured materials and channels for micro- or nanofluidics. Regarding the recently flourishing research realm of living or self-motile particles, a challenging extension of the present work could consist of systematically studying the influence of activity on the predicted equilibrium state diagram[62]. Finally, a fascinating connection with biology emerges from drawing the analogies between colloidal liquid crystals and growing colonies of rod-shaped bacteria[63–66]. Our results thus lay the foundation for a deeper microscopic understanding of the structures emerging and persisting along with the evolution dynamics when such living systems are subjected to extreme topological confinement.

## Methods

**Sedimentation of silica rods**. To experimentally create confined quasi-two-dimensional smectic structures, we take advantage of the phase stacking of silica rods[37,38] in sedimentation equilibrium[39]. The bare dimensions of the rods are measured directly from scanning electron microscopy images. The rods are dispersed into a 1 mM NaCl water solution to ensure stability through double-layer repulsion. Introducing effective dimensions (see Supplementary Note 1) to account for the Debye screening, our particles behave like hard rods of an effective aspect ratio $p_{eff} = 10.6$.

The confining cavities (see Fig. 1a) in the shape of hollow cylinders are molded on the bottom coverslip using home made Polydimethylsiloxane (PDMS) stamps and Norland Optical Adhesive[39]. In practice, several chambers are fitted in a single cell. After preparation, the rod solution is left in the tube to sediment for at least 12 h. During sedimentation, the concentration of particles gradually increases along the direction of the gravity field leading to successive isotropic, nematic and smectic order at the bottom (see Supplementary Movie 1). After a few hours, sedimentation diffusion equilibrium is reached and the three phases coexist in the cavities.

The smectic structures are observed by means of bright-field microscopy in the direct vicinity of the bottom wall. We use a 1.42 numerical aperture apochromat oil immersion objective mounted on an Olympus IX73 microscope and coupled to a Ximea CMOS xiQ camera, which allows an optical resolution comparable to the rod diameter. Due to degenerate planar anchoring at the bottom wall, the system can be considered a quasi-two-dimensional fluid in annular confinement. We choose the total amount of rods to obtain an effective volume fraction $\phi_{eff} \approx 45 - 50\%$ close to the bottom. This ensures that there is no crystalline state and that the rods in direct contact with the bottom wall exhibit smectic order. To create some statistics, we repeat the measurements in a given geometry up to 12 times.

**Density functional theory (DFT)**. We study by free minimization of a DFT[42] in two dimensions hard discorectangles (see Fig. 1c) with an aspect ratio $p = 10$ that well reflects the experimental parameters. The interaction between these particles is described by a free energy functional constructed as an extension of fundamental measure theory[43,67,68] to account for anisotropic particle shapes[45,69]. These geometrical functionals derived from first principles are exact in the low-density limit and have proven very reliable for highly packed systems. The annular confinement is included as an external hard-wall potential.

The key quantity in our theory is the one-body density profile $\rho(\mathbf{r}, \phi)$, providing the probability to find a particle with the center-of-mass position $\mathbf{r}$ and its symmetry axis oriented along an angle $\phi$. Consider now a density functional $\Omega[\rho(\mathbf{r}, \phi)] = \mathcal{F}[\rho] + \int d\mathbf{r} \int_0^{2\pi} \frac{d\phi}{2\pi} \rho(\mathbf{r}, \phi)(V_{ext}(\mathbf{r}, \phi) - \mu)$, where $\mathcal{F}[\rho]$ is the intrinsic Helmholtz free energy functional, $V_{ext}(\mathbf{r}, \phi)$ is the external potential

and $\mu$ the chemical potential (more details can be found in Supplementary Note 1). Then the density $\rho(\mathbf{r}, \phi)$ of a (meta-) stable state is found by iteratively solving the extremal condition $\delta\Omega[\rho]/\delta\rho = 0$ starting from a particular initial guess for the density profile. The average area fraction $\eta = 0.65$ is kept fixed throughout the iteration by adapting $\mu$ in each step. Then we compare the values of the free energy $\mathcal{F}[\rho]$ of the different structures to identify the global minimum and quantify the likelihood to observe a metastable local minimum $\rho(\mathbf{r}, \phi)$ in a corresponding experiment. Calculations are performed on a quadratic spatial grid with a high enough resolution $\Delta x = \Delta y = 0.2$ and $N_\phi = 96$ discrete orientational angles. We iterate until the free energy differences between different structures can be sufficiently resolved.

**Overlapping parameter space**. Our experiment and theory are designed, such that they can both tackle hard rods with a comparable anisotropy that is sufficiently high to ensure that the smectic phase is stable over a large range of densities[45,70]. Systems with variable inclusion size ratios $b$ and the radii $R_{out}$ of the circular outer wall ranging between $1.9L \leq R_{out} \leq 5.7L$ are covered by both approaches.

**Data analysis and presentation**. From the theoretical data, we determine a local packing fraction by weighting the density with the local particle area to highlight the particle resolution within our data (second row of Fig. 2). As a standard representation of the full density field we use in the third row of Fig. 2 and for all other illustrations a plot of the orientationally averaged density with the orientational director field, represented by green arrows of length given by the local order parameter. Moreover, we also directly display the local order parameter field (fourth row of Fig. 2). Note that the distorted appearance of the order parameter close to the inclusion in the laminar state reflects the very low but nonvanishing probability to find particles left and right of the symmetry axis that are perfectly aligned with the wall. This underlines the similarity of interior and boundary defects illustrated in Fig. 5. We compare the free energies of the different structures in Figs. 3 and 6. Further details on numerical errors are given in Supplementary Note 1.

The experimental snapshots (fifth row of Fig. 2) are inspected visually and further processed using Wolfram Mathematica computing system. This allows us to color each particle according to its orientation relative to the wall (sixth row of Fig. 2). From the measured center-of-mass positions and orientations we further extract a local field of the orientational order parameter (seventh row of Fig. 2).

The different smectic states and their microscopic structures are identified according to the criteria described in Supplementary Note 2. Further details and a collection of the raw data can be found in Supplementary Note 8.

## Data availability

All theoretical structures from which some data were extracted and discussed in this work are collected in Supplementary Figs. 6–15. The collected experimental bright-field images before and after processing are shown in Supplementary Figs. 16–25. See Supplementary Data 1 for full-size versions of these Supplementary Figures and Supplementary Data 2 for the raw images and a manual for image processing.

Source data are provided with this paper. Further data supporting the findings of this study are available from the corresponding authors upon reasonable request.

## Code availability

An example program to minimize the density functional is provided as Supplementary Software 1. Additional software used in this study is available from the corresponding authors upon reasonable request.

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

## Acknowledgements

We thank Christoph E. Sitta for implementing large parts of the DFT code, Paul A. Monderkamp for helpful discussions and providing a plotting tool, and Axel Voigt for helpful discussions. This work was supported by the German Research Foundation (DFG) within project LO 418/20-2. This project has received funding from the European Union's Horizon 2020 research and innovation programme under the Marie Skłodowska-Curie Grant Agreement No. 641839.

## Author contributions

L.B.G.C. and D.G.A.L.A. designed the experiments and L.B.G.C. performed the experiments. R.W. designed and performed the density functional study. All authors analyzed and interpreted the experimental and theoretical data. R.W. and L.B.G.C. processed the data and created the figures. R.W., L.B.G.C., and H.L. wrote the manuscript.

## Funding

## Competing interests

The authors declare no competing interests.
