## [Peer Review File · Nature Communications]

REVIEWER COMMENTS

Reviewer #1 (Remarks to the Author):

This work reports on the formation of topological defects in smectic lyotropic liquid crystals in annular confinement. Both theoretical results based on density functional theory and experimental results of confined colloidal rods are presented and agree with each other. Several smectic-like states are formed as a result of a delicate balance between elastic deformations of both the director field and the smectic period, surface energies, and the energy associated to the formation of topological defects. Experimental results are collected in a state diagram which is also supported by theoretical free-energy calculations of the relative stability of each state. The Shubnikov and the composite states are new. The stability of the composite states is intriguing.

Overall, the work seems to be carefully done, it is new (studies of confined smectic states are uncommon and typically limited to planar geometries), and will likely influence researchers in the field. I would like the authors to address the following points before recommending the paper for publication.

A) The dimensions of the density in Fig. 1 are not specified. Only in the supplementary Eq. (6) a dimensionless scaled density is introduced which may or may not be the same as the one used in the main text. Also, a color bar for the color-coded density profiles is missing in most figures (only included in Fig. 1). The color bar for second row of Fig. 2 (local packing fraction) is not presented either.

B) It would help if the aspect ratio of the particles and the overall packing fraction in both experiments and theory are given in the main text (not only in the methods and supplementary information) since they play a crucial role in lyotropic liquid crystals. In line with this comment, I could not find an estimate of the experimental packing fraction (or number density). Is the theoretical packing fraction selected to match the experimental packing fraction?

C) Liquid crystals in both circular and annular confinement, have been considered before in the limit of much lower density (relevant studies are already cited in the introduction). Briefly discussing how the high density smectic-like states presented in e.g. Fig. 3 might be connected to the "low" density nematic-states by decreasing the overall density (e.g. via a sharp transition or rather a continuous deformation of the director field) would be interesting and it would help to contextualize the work. The authors mention that isotropic and nematic states are formed during the sedimentation process (increasing density). Are the observed nematic states the equilibrium states if the system were equilibrated at that precise density? If the answer is positive, a single run of experiments could be used to obtain a complete state diagram similar to that in Fig. 3 but with a third axis that measures the overall density.

D) In a continuous description, the Euler characteristic determines the total topological charge inside the cavity (see e.g. Bowick, Mark J., and Luca Giomi. "Two-dimensional matter: order, curvature and defects." *Advances in Physics* 58.5 (2009) 449-563). This does not seem to be the case here since different states possess different total topological charge inside of the cavity, even though the cavity has the same Euler characteristic. How do you explain that the Euler theorem does not hold here?

E) In line with the previous point, in large cavities the laminar state dominates for small inclusions and the Shubnikov dominates for large inclusions. However, a continuous description should be valid at some point by

just increasing the size of the system (R_{out}). Since the Shubnikov and the laminar states have different topological charge, the state diagram (Fig. 3) should change for (very) large cavities. What is the expected state diagram for cavities much larger than those considered here?

F) I could not find information about the relative orientational order of the different states. Plots of the uniaxial orientational order parameter would be valuable to compare the relative orientational order between the different states and to improve the visualization of topological defects. For example, the second (theory) and fourth (experiments) rows of Fig. 2 could present maps of the orientational order parameter instead of the local packing fraction (second row) and grey particles (fourth row) which do not add much to the figure.

G) What is the expected effect of changing the aspect ratio of the particles? In a closely related work, smectic defects in annular confinement have been recently considered in Ref. [40] for the case of granular particles of relatively low length-to-width aspect ratio. The authors of [40] find "tetratic"-like states. Do you expect large changes in the state diagram by changing the aspect ratio of the rods? Can the aspect ratio of the rods be tuned in the experiments?

~

Reviewer #2 (Remarks to the Author):

The topological frustrations created by boundary conditions on an ordered liquid have been an interesting problem in physics, mathematics and materials science. In this manuscript, the authors examine the defect patterns of a smectic liquid crystal confined inside annular circles, which display quite a number of exotic defect states. The approaches they use, combining experimental measurement and theoretical calculation, have convincingly shown the existence of these eye-opening defect states. This is a pioneering study. The manuscript is well presented and I recommend its publication. I only have one minor comment.

To help the readers easily compare the theoretical and experimental results, the DFT images in Figs. 2 and 4 could be rotated to match the orientation of the patterns from the experimental image. Currently, they are at different orientational angles.

Reviewer #3 (Remarks to the Author):

The present manuscript discusses defect topologies that arise in smectic liquid-crystalline materials under severe confinement conditions of an annular shape. On the experimental side the investigation is conducted by real-space microscopy whereas the theoretical part employs density functional theory based upon Rosenfeld's fundamental measure theory for hard-body interactions. The systems have been very carefully selected so that both theory and experiment are located in a sufficiently overlapping parameter space. I was very much impressed with the similarity of characteristic defect topologies observed by both approaches.

These defect topologies are referred to as laminar, domain, and so-called Shubnikov. Their characteristics are clearly explained and illustrated in Fig. 2 both from the experimental and from the theoretical side. A fourth, so-called composite state, combines several characteristics of the other three and is apparently not considered

a fourth class of states.

In my opinion this work is a rare example of a very interesting and successful experimental/theoretical study presenting novel results that could potentially be useful in the directed self-assembly of materials in complex fluids. The very impressive similarity of the very rich and complex structures that are observed in this work can be seen in Fig. 4 of the manuscript. This richness applies also to the "phase diagram" plotted in Fig. 3 in the inclusion radius/relative inclusion size plane. It is further illustrated by the free-energy plot in Fig. 5. In conclusion, Overall this work is of high quality and the conclusions are very well supported by the material presented. I expect this study to be of broad interest and I am therefore happy to recommend its publication as is.

Martin Schoen

Detailed response the first reviewer

Reviewer #1 (Remarks to the Author):

This work reports on the formation of topological defects in smectic lyotropic liquid crystals in annular confinement. Both theoretical results based on density functional theory and experimental results of confined colloidal rods are presented and agree with each other. Several smectic-like states are formed as a result of a delicate balance between elastic deformations of both the director field and the smectic period, surface energies, and the energy associated to the formation of topological defects. Experimental results are collected in a state diagram which is also supported by theoretical free-energy calculations of the relative stability of each state. The Shubnikov and the composite states are new. The stability of the composite states is intriguing.

Overall, the work seems to be carefully done, it is new (studies of confined smectic states are uncommon and typically limited to planar geometries), and will likely influence researchers in the field. I would like the authors to address the following points before recommending the paper for publication.

We thank the reviewer for the careful reading and recommending our manuscript for publication after revision. We find that all points are important and considerably modified our manuscript accordingly. Please find below our detailed response:

A) The dimensions of the density in Fig. 1 are not specified. Only in the supplementary Eq. (6) a dimensionless scaled density is introduced which may or may not be the same as the one used in the main text. Also, a color bar for the color-coded density profiles is missing in most figures (only included in Fig. 1). The color bar for second row of Fig. 2 (local packing fraction) is not presented either.

The reviewer is right that the definition of the plotted density was given only in Supplementary Eq. (6). In our revised manuscript, the definitions of particle area and normalized orientationally averaged center-of-mass density were added to the caption of Fig. 1.

We also added color bars to all rows of Fig. 2 and wrote some additional legends to the right panel. By doing so, we shortened the caption of Fig. 2 without sacrificing important information. Regarding the density plots in Figs. 4, 6 and 7 (and some Supplementary figures), we added some comments like “as in Fig. 1d” to the caption, when referring to the plot appearance. For the new Supplementary Fig. 5, which uses a different color bar (which is explicitly shown), we explicitly wrote “Note that the color bar only has half the range as in the all other density plots throughout the Supplementary figures and the main manuscript”

B) It would help if the aspect ratio of the particles and the overall packing fraction in both experiments and theory are given in the main text (not only in the methods and supplementary information) since they play a crucial role in lyotropic liquid crystals. In line with this comment, I could not find an estimate of the experimental packing fraction (or number density). Is the theoretical packing fraction selected to match the experimental packing fraction?

The experimental and theoretical aspect ratios are now also stated in the overview section of the main manuscript, stating that “The theoretical aspect ratio $p = 10$ is chosen to closely match the effective value $p_{\text{eff}} = 10.6$ in the experiment”. The theoretical value $p = 10$ was added to Fig. 1c

We also added the comment “the density of rods is chosen to be slightly above the bulk nematic–smectic transition in each case” to the overview section and further elaborate the necessity for such a relative choice in Supplementary Note 1: “As this two-dimensional area fraction is not directly comparable to the experimentally accessible volume fraction ϕ , our theoretical calculations in the confined system are generally carried out for $\eta = 0.65$. With this choice of the smectic density, the relative deviation from the bulk nematic–smectic transition density in experiment and theory are similar.” Moreover, the effective experimental volume fraction “ $\phi_{\text{eff}} \approx 45 - 50\%$ close to the bottom” is now also stated in the methods section (before it was only noted in Supplementary Note 1 as pointed out by the reviewer).

C) Liquid crystals in both circular and annular confinement, have been considered before in the limit of much lower density (relevant studies are already cited in the introduction). Briefly discussing how the high density smectic-like states presented in e.g. Fig. 3 might be connected to the “low” density nematic-states by decreasing the overall density (e.g. via a sharp transition or rather a continuous deformation of the director field) would be interesting and it would help to contextualize the work. The authors mention that isotropic and nematic states are formed during the sedimentation process (increasing density). Are the observed nematic states the equilibrium states if the system were equilibrated at that precise density? If the answer is positive, a single run of experiments could be

used to obtain a complete state diagram similar to that in Fig.3 but with a third axis that measures the overall density.

The reviewer raises some interesting points, which we now elaborate briefly in the new section “Dependence on density and rod length” of our manuscript and further detail in Supplementary Note 7. To support this discussion (and the one on point G)), we moved Fig. 7 to the main manuscript to illustrate that the stability of the observed structures indeed changes when the density is varied. We further newly created Supplementary Fig. 4 on an unstable smectic structure that is related to a nematic state with threefold symmetry, Supplementary Fig. 5 on the theoretical nematic–smectic transition in confinement and Supplementary video 1 showing the formation of nematic states in course of the sedimentation process in the experiment.

To briefly answer the reviewer’s explicit questions we make here the following remarks (which are included in the manuscript as part of the new section “Dependence on density and rod length”). Smectic and nematic states can be connected to each other regarding the director topology, while the shape of the topological defects is considerably different (see also our new discussion of topological details as response to point D)). Hence the transition is not continuous. In sedimentation equilibrium, the nematic structures are probably influenced by the smectic ones below, as these are more rigid.

D) In a continuous description, the Euler characteristic determines the total topological charge inside the cavity (see e.g. Bowick, Mark J., and Luca Giomi. "Two-dimensional matter: order, curvature and defects." Advances in Physics 58.5 (2009) 449-563). This does not seem to be the case here since different states possess different total topological charge inside of the cavity, even though the cavity has the same Euler characteristic. How do you explain that the Euler theorem does not hold here?

We are very grateful that the reviewer made this comment, which helped us a lot to improve our manuscript. This seeming contradiction is probably the result of a too compact account of the peculiar topological properties of our observed states. By the winding number k we did not mean the charge of a topological defect within the director field, but rather aimed to quantify the director alignment around the inclusion in a simple way. In the language of topological charge within the fluid, misalignment at the inclusion should be interpreted as a $-1/2$ defect whenever it occurs.

In both cases, the Euler theorem holds for all states, but it has to be considered in a different way, depending on whether or not the inclusion is considered (a) as a hole or (b) formally as part of the fluid with own defect charge. Case (a) corresponds to the “standard” conservation law that the Euler characteristic χ equals the total topological charge Q . In case (b), the modified form $\chi = \tilde{Q} + k - h$ holds, where \tilde{Q} is the total charge without including the misalignment at the inclusion and h is the number of holes.

We understand that this may have caused confusion and made a major revision of the related parts of our manuscript as follows.

- We removed the notion of the winding number k around the inclusion completely from the main text, deferring the more elaborate discussion of case (b) to Supplementary note 5, where

we also discuss the implications of charge conservation. In particular, we emphasize that there is no violation of the Euler theorem.

- In the main text, we rather only speak of topological defects, i.e., case (a). This turned out to be particularly interesting, since we could associate different topological charges with the different line defects, which either appear within the system or directly at the boundary. Taking a closer look at the microscopic structure allows for the interpretation that the defect charge sits at the two end points of a defect line. These intriguing issues are elaborated in the new section “Topological details” and schematically illustrated in the new Figure 5.
- The reference mentioned by the reviewer has been included as Ref. [46] and cited upon stating that the total charge Q “equals the Euler characteristic of the confining domain, as required by topology”.
- A statement at the beginning of the paragraph “smectic states” has been extended mentioning “topological defects with total charge of $Q = \chi = 0$ (see topological details below)” to emphasize already at this point in the manuscript that the Euler theorem holds in general.
- Some parts from the section “Microscopic details” discussing the topology of the domain states were moved to the new section “Topological details”
- We included the additional word “topological” in the title of our manuscript, since this major revision led us to put a stronger emphasis on the concept of topology.

E) In line with the previous point, in large cavities the laminar state dominates for small inclusions and the Shubnikov dominates for large inclusions. However, a continuous description should be valid at some point by just increasing the size of the system (R_{out}). Since the Shubnikov and the laminar states have different topological charge, the state diagram (Fig. 3) should change for (very) large cavities. What is the expected state diagram for cavities much larger than those considered here?

This point is probably a consequence of the misconception, which we hopefully clarified in the detailed response to point D) - the competition between both states should still exist in the continuum limit. We added the comment “For large confinements, the transition seems to approach the continuum limit with $b \approx 0.27$ ” regarding the Laminar–Shubnikov transition to the discussion of the state diagram.

F) I could not find information about the relative orientational order of the different states. Plots of the uniaxial orientational order parameter would be valuable to compare the relative orientational order between the different states and to improve the visualization of topological defects. For example, the second (theory) and fourth (experiments) rows of Fig. 2 could present maps of the orientational order parameter instead of the local packing fraction (second row) and grey particles (fourth row) which do not add much to the figure.

We thank the reviewer for making the excellent suggestion of plotting the orientational order parameter, which very much aids the discussion of the topological defects. We added the new order-parameter plots to Fig. 2 and adapted the paragraph “Data analysis and presentation” in

the methods section accordingly. We also extended the statement in Supplementary Note 1 on how the order parameter and director field are generally obtained from DFT.

G) What is the expected effect of changing the aspect ratio of the particles? In a closely related work, smectic defects in annular confinement have been recently considered in Ref. [40] for the case of granular particles of relatively low length-to-width aspect ratio. The authors of [40] find "tetratic"-like states. Do you expect large changes in the state diagram by changing the aspect ratio of the rods? Can the aspect ratio of the rods be tuned in the experiments?

Together with the elaborations on point C), these issues are discussed in the new section "Dependence on density and rod length" and illustrated in Fig. 7b. To answer the reviewer's question, we explicitly wrote "Extending our state diagram towards shorter rods at a fixed density, we further anticipate the emergence of stable tetratic structures [41] (this was Ref. [40]), since smectic order is generally destabilized [47]." The new reference [47] shall illustrate that tetratic order can be described with DFT and the mentioned destabilization of smectic order, which is the reason why we focused on larger rods.

In short, it is possible that the described smectic states become metastable with respect to states with tetratic order. This could, in principle, be studied by synthesizing shorter rods in the experiment.

Response the second reviewer

Reviewer #2 (Remarks to the Author):

The topological frustrations created by boundary conditions on an ordered liquid have been an interesting problem in physics, mathematics and materials science. In this manuscript, the authors examine the defect patterns of a smectic liquid crystal confined inside annular circles, which display quite a number of exotic defect states. The approaches they use, combining experimental measurement and theoretical calculation, have convincingly shown the existence of these eye-opening defect states. This is a pioneering study. The manuscript is well presented and I recommend its publication. I only have one minor comment.

To help the readers easily compare the theoretical and experimental results, the DFT images in Figs. 2 and 4 could be rotated to match the orientation of the patterns from the experimental image. Currently, they are at different orientational angles.

We thank the reviewer for the appreciation of our work and the recommendation for publication. The DFT plots in Figs. 2 and 4 have been rotated as suggested.

Response the third reviewer

Reviewer #3 (Remarks to the Author):

The present manuscript discusses defect topologies that arise in smectic liquid-crystalline materials under severe confinement conditions of an annular shape. On the experimental side the investigation is conducted by real-space microscopy whereas the theoretical part employs density functional theory based upon Rosenfeld's fundamental measure theory for hard-body interactions. The systems have been very carefully selected so that both theory and experiment are located in a sufficiently overlapping parameter space. I was very much impressed with the similarity of characteristic defect topologies observed by both approaches.

These defect topologies are referred to as laminar, domain, and so-called Shubnikov. Their characteristics are clearly explained and illustrated in Fig. 2 both from the experimental and from the theoretical side. A fourth, so-called composite state, combines several characteristics of the other three and is apparently not considered a fourth class of states.

In my opinion this work is a rare example of a very interesting and successful experimental/theoretical study presenting novel results that could potentially be useful in the directed self-assembly of materials in complex fluids. The very impressive similarity of the very rich and complex structures that are observed in this work can be seen in Fig. 4 of the manuscript. This richness applies also to the "phase diagram" plotted in Fig. 3 in the inclusion radius/relative inclusion size plane. It is further illustrated by the free-energy plot in Fig. 5. In conclusion, Overall this work is of high quality and the conclusions are very well supported by the material presented. I expect this study to be of broad interest and I am therefore happy to recommend its publication as is.

Martin Schoen

We thank the reviewer for the appreciation of our work and the recommendation for publication.

REVIEWERS' COMMENTS

Reviewer #1 (Remarks to the Author):

The authors have addressed my comments. I consider this revised version suitable for publication.

Daniel de las Heras

Response the first reviewer

Reviewer #1 (Remarks to the Author):

The authors have addressed my comments. I consider this revised version suitable for publication.

Daniel de las Heras

We thank the reviewer again for the helpful comments from the previous report and the recommendation for publication.